# Integrated Proteomics and Lipidomics Investigation of the Mechanism Underlying the Neuroprotective Effect of *N*-benzylhexadecanamide

**DOI:** 10.3390/molecules23112929

**Published:** 2018-11-09

**Authors:** Yanyan Zhou, Hongjie Wang, Feifei Guo, Nan Si, Adelheid Brantner, Jian Yang, Lingyu Han, Xiaolu Wei, Haiyu Zhao, Baolin Bian

**Affiliations:** 1Institute of Chinese Materia Medica, China Academy of Chinese Medical Sciences, Beijing 100700, China; yyzhou@icmm.ac.cn (Y.Z.); hjwang@icmm.ac.cn (H.W.); ffguo@icmm.ac.cn (F.G.); nsi@icmm.ac.cn (N.S.); jyang@icmm.ac.cn (J.Y.); hanlingyu1990@foxmail.com (L.H.); xlwei@icmm.ac.cn (X.W.); 2Institute of Pharmaceutical Sciences, University of Graz, Graz 8010, Austria; adelheid.brantner@uni-graz.at

**Keywords:** *N*-benzylhexadecanamide, neuroprotective effect, neurotransmitter, proteomics, lipidomics

## Abstract

Macamides are very important secondary metabolites produced by *Lepidium meyenii* Walp, which possess multiple bioactivities, especially in the neuronal system. In a previous study, we observed that macamides exhibited excellent effects in the recovery of injured nerves after 1-methyl-4-phenylpyridinium (MPP^+^)-induced dopaminergic neuronal damage in zebrafish. However, the mechanism underlying this effect remains unclear. In the present study, we observed that *N*-benzylhexadecanamide (XA), which is a typical constituent of macamides, improved the survival rate of neurons in vitro. We determined the concentration of neurotransmitters in MN9D cells and used it in conjunction with an integrated proteomics and lipidomics approach to investigate the mechanism underlying the neuroprotective effects of XA in an MPP^+^-induced neurodegeneration cell model using QqQ MS, Q-TOF MS, and Orbitrap MS. The statistical analysis of the results led to the identification of differentially-expressed biomarkers, including 11 proteins and 22 lipids, which may be responsible for the neuron-related activities of XA. All these potential biomarkers were closely related to the pathogenesis of neurodegenerative diseases, and their levels approached those in the normal group after treatment with XA. Furthermore, seven lipids, including five phosphatidylcholines, one lysophosphatidylcholine, and one phosphatidylethanolamine, were verified by a relative quantitative approach. Moreover, four proteins (Scarb2, Csnk2a2, Vti1b, and Bnip2) were validated by ELISA. The neurotransmitters taurine and norepinephrine, and the cholinergic constituents, correlated closely with the neuroprotective effects of XA. Finally, the protein–lipid interaction network was analyzed. Based on our results, the regulation of sphingolipid metabolism and mitochondrial function were determined to be the main mechanisms underlying the neuroprotective effect of XA. The present study should help us to better understand the multiple effects of macamides and their use in neurodegenerative diseases.

## 1. Introduction

Macamides are representative lipophilic constituents of *Lepidium meyenii* (Maca), with various promising pharmacological properties, including neuroprotective, anti-fatigue, and fertility improving effects [1,2,3,4,5]. In previous reports, they were demonstrated to act on CB1 receptor for their neuroprotective activity against Mn-induced mitochondrial depolarization and toxicity in U-87 cells [5]. In addition, the inhibition of fatty acid amide hydrolase (FAAH) was regarded as another mechanism for the anti-inflammatory and neuroprotective effects of macamides [6]. In a previous study, we also observed excellent effects of macamides on dopaminergic neuronal repair in 1-methyl-4-phenylpyridinium (MPP^+^)-induced zebrafish model [7]. *N*-benzylhexadecanamide (XA) is one of the most abundant ingredients among macamides. XA was demonstrated to exhibit neuroprotective effects in the zebrafish model. An in vitro evaluation using AChE and BuChE inhibition assays also supported this result [7]. 

The MPP^+^-induced MN9D cell line is a classical model for neuronal injury [8]. MPP^+^ causes disorders in sphingolipid metabolism, tyrosine metabolism, and mitochondrial function [9]. The neurotoxin MPP^+^ faithfully replicates the biological and pathological hallmarks of neurodegenerative diseases [10]. Fortunately, the molecular mechanisms underlying the injury caused by MPP^+^ may corroborate with those regarded to be involved in the neuroprotective effects of macamides [7]; therefore, this cell model can be used for assaying the effects of macamides.

The concentration of neurotransmitters reflects the status of the central nervous system [11,12,13]. In recent years, ultra-high performance liquid chromatography coupled with triple quadrupole mass spectrometry (UPLC-QqQ-MS/MS) has been used for the comprehensive determination of neurotransmitters, with high sensitivity and efficiency [14]. 

Integrated omics-platforms have allowed the investigation of pathways and screening of potential biomarkers, which have provided deeper insights into the mechanisms of diseases [15,16]. Advanced liquid chromatography-mass spectrometry (LC-MS) technology offers an important technical support. Simultaneously, bioinformatics tools allow integration between different biomarkers, such as lipids and proteins. To our knowledge, there are still no reports about the integration of lipidomics and proteomics in the field of research on neurodegenerative diseases. 

Lipidomics provides important insights into the biochemical mechanisms for evaluation of the neuroprotective effects of compounds. Potential lipid markers have been reported previously [17,18,19]. For example, the levels of cholesterol ester, cholesterol, cerebroside, and phosphatidylcholine in the brain were closely related to neurodegenerative diseases [20]. With regard to proteomics, the potential biomarkers contributed greatly to the elucidation of new mechanisms for the patho-physiology of neurodegenerative diseases [21,22]. The analysis of protein–protein interactions revealed a neuroprotective effect of huperzine A against oligomer-induced cell death through the regulation of p53 (Trp53) [23]. Also, molecular targets of granulocyte colony stimulating factor in the mouse brain and PC12 cell line were discovered by proteomic analysis [24].

In the present study, the concentration of 27 neurotransmitters in MN9D cells and cell supernatant, before and after administration of XA, were measured by UPLC-QqQ MS/MS. Taurine, norepinephrine, and cholinergic constituents displayed close association with neuroprotective effects of XA. Based on the Q-TOF MS and Orbitrap MS technology, the lipidomic and proteomic study of XA on MPP+-induced MN9D cells were carried out. We identified 11 proteins and 22 lipids, which were closely related to the pathogenesis of neurodegenerative diseases. Furthermore, differential expression of seven lipids [five phosphatidylcholines (PCs), one lysophosphatidylcholine (LysoPC), one phosphatidylethanolamine (PE)], and four proteins (Scarb2, Csnk2a2, Vti1b, and Bnip2) was verified using a relative quantitative approach and ELISA test, respectively. The regulation of sphingolipid metabolism and mitochondrial function might be the underlying mechanism for the neuroprotective effect of XA.

## 2. Material and Methods

### 2.1. Materials

XA was isolated from Maca by the authors (purity ≥ 95.0%). The 95% ethanol extract of L. *meyenii* tubers was subjected to column chromatography (CC) for coarse division and subdivision, followed by semi-preparative liquid chromatography to yield compound XA. A purity analysis was carried out by LC-MS and NMR. The data from LC-MS, ^1^H-NMR and ^13^C-NMR was identified as *N*-benzylhexadecamide (XA) by comparison with the literature [25]. 

MPP^+^ was purchased from Sigma Aldrich (Steinheim, Germany). The reference compounds for aspartic acid, asparagine, serine, taurine, tyrosine, noradrenaline, homovanillic acid, choline, acetylcholine, butyrylcholine, ornithine, lysine, phenylalanine, tryptophan, l-leucine, methionine, dimethylglycine, proline, histamine, threonine, citrulline, arginine, serotonin, adrenaline, dopamine, levodopa, and γ-aminobutyric acid and the internal standard, diazepam, were purchased from National Institute for Food and Drug Control (Beijing, China).

### 2.2. Cell Culture

MN9D cells, derived from dopaminergic neurons in the midbrain of mice, were purchased from Shanghai Chen Biotechnology Co., Ltd. RPMI-1640 medium, EDTA-trypsin (0.25%), fetal bovine serum (FBS), penicillin (100 U/mL), streptomycin (100 mg/mL), and phosphate-buffered saline (PBS) were bought from Gibco (Grand Island, NY, USA). The cells were cultured in 1640 medium, supplemented with 10% FBS and 1% penicillin/streptomycin, at 37 °C in a humidified atmosphere with 5% CO_2_.

### 2.3. Cell Viability Assay

For cell viability assays, MN9D cells were seeded in 96-well plates at a density of 8000 cells/well. After incubation for 24 h at 37 °C, the cells were treated with different concentrations of MPP^+^ (67.5–1000 μM) for 24 h to mimic the MPP^+^-induced neuron damage model; then, the medium was added for 48 h. The CCK-8 reagent was used to assess cell viability. After incubation of optimal MPP^+^ concentration for 24 h at 37 °C, the cells were treated with different concentrations of XA (12.5–200 μM) for 48 h. Cell viability was determined by CCK-8 assay according to a previously-described method, with slight modifications [26]. The absorbance was measured at 450 nm using a microplate reader (Varioskan Flash Microplate Reader, Thermo Corporation, USA). The percentage survival was calculated relative to that in the control wells in which only the RPMI-1640 medium and DMSO was added (the survival in the control wells was considered to be 100%). The vehicle (DMSO) should be less than 0.1%.

### 2.4. Observation of Cell Morphology

The cells were exposed to different treatments as per the experimental group. The medium and DMSO was added in the control group. As for model group, the cells were treated with optimized concentrations of MPP^+^ for 24 h. Then, the medium was added for 48 h. For the administration group, after incubation for 24 h at 37 °C, the cells were treated with the optimized concentrations of MPP^+^ for 24 h. Then, the cells were treated with the optimized concentration of XA for 48 h. The morphology of the cells in each group was observed under an inverted microscope (CKX41 OLYMPUS Inverted Microscope, Olympus Corporation, Tokyo, Japan), and the cells were photographed.

### 2.5. Measurement of Neurotransmitters 

#### 2.5.1. Preparation of Standards Solutions. 

The reference standards were dissolved in 50% HPLC grade methanol to obtain a stock solution with a concentration of 1.0 mg/mL for each neurotransmitter. The stock solution of the internal standard diazepam was prepared at a concentration of 45.545 µg/mL. The standard stock solutions were kept at 4 °C before analysis.

#### 2.5.2. Pretreatment of Cell Supernatants and Cell Samples. 

The cells were incubated in an atmosphere of 5% CO_2_ at 37 °C. The extracellular concentration of the neurotransmitters was determined in the collected medium. The medium from each dish was pipetted into a 15 mL centrifuge tube and centrifuged at 179× *g* for 5 min. Thereafter, the supernatant was transferred to a 10 mL eppendorf tube and centrifuged at 12,000 × *g* for 10 min at 4 °C. The supernatant was stored at −20 °C until analysis. When analyzed, 20 μL internal standard solution was added to 200 μL of the cell supernatant.

To determine the intracellular concentration of the analytes, the adherent cells were washed twice with 4 mL cold PBS, and 0.5 mL trypsin was added to each dish. The dislodged cells were centrifuged at 179× *g* for 5 min at 4 °C, after which the supernatant was discarded. The cell pellet was thawed, and the cells were lysed with an ultrasonic cell disruptor after the addition of 2 mL water containing 0.2% formic acid, followed by precipitation and centrifugation. When analyzed, 20 μL internal standard solution was added to 200 μL of cell supernatant [27].

#### 2.5.3. Chromatographic Conditions. 

An Agilent 6490A triple quadrupole LC-MS system (Agilent Corporation, Palo Alto, MA, USA), equipped with G1311A quaternary pump, G1322A vacuum degasser, G1329A autosampler, and G1316A thermostat, was employed. The separation was achieved at 25 °C using a Waters ACQUITY UPLC BEH Amide column (2.1 mm × 100 mm, 1.7 µm). The mobile phase consisted of water containing 0.2% formic acid (A) and acetonitrile (B), and was used at a flow rate of 0.3 mL/min. The gradient program used was as follows: 90% B from 0 to 1 min, 90–75% B from 1 to 10 min, and 75–50% B from 10 to 13 min. The injection volume was 1 μL.

The analytes were determined by monitoring the precursor–product transition in MRM mode using ion polarity switching mode. To ensure the desired abundance of each compound, the CE values and other parameters were optimized and were as follows: cycle time, 300 ms; gas temp, 200 °C; gas flow, 14 L/min; nebulizer, 20 psi; sheath gas flow, 11 L/min; capillary voltage, 3 kV; nozzle voltage, 1.5 kV; Delta EMV (+), 200 V. The optimized mass transition ion pairs (*m*/*z*) and CE values for neurotransmitters are shown in Table 1. The MRM chromatograms of 10 neurotransmitters are shown in Figure 1.

## 3. Lipidomic and Proteomic Analyses

### 3.1. Incubation Conditions for Lipidomic and Proteomic Studies. 

After treatment with trypsin, the cells were plated at a density of 5 × 10^6^ cells/dish. The cells were cultured for 24 h and divided into the control, model (500 µM MPP^+^), and administration (100 and 50 µM XA) groups.

### 3.2. Preparation of Lipid Samples.

After treatment with XA and incubation at 37 °C under 5% CO_2_ for 48 h, the cells were washed five times with cold PBS containing 1 mM EDTA. The cells were then frozen in liquid nitrogen, collected with a cell scraper, and transferred to a glass centrifuge tube. Thereafter, 3.75 mL of a precooled mixture of dichloromethane and methanol (1:2) was added and the suspension was vortexed for 1 min. The liquid was then frozen on ice for 30 min and 1.25 mL dichloromethane was added to it; the mixture was vortexed for 1 min. Pure water (1.25 mL) was then added and the solution was again vortexed for 5 min and centrifuged at 10,000× *g* at 4 °C for 10 min. The dichloromethane layer (lower part) was transferred to a new glass tube. The extraction process was repeated twice. The dichloromethane layers were combined, dried under a nitrogen blow dryer, and stored at −80 °C. The dried residue was reconstituted with the methanol–dichloromethane mixture (1:1, containing 5 mM ammonium acetate) for MS analysis.

### 3.3. Preparation of Protein Samples

The proteins were extracted from MN9D cells with 8 M urea on ice bath, and 300 μg protein (in a volume of 44.79 μL) was reduced by adding 4.98 μL of 0.1 M dithiothreitol and incubating at 37 °C for 4 h. The proteins were then alkylated by adding 5.53 μL of 0.5 M iodoacetamide and incubating in the dark at 25 °C for 60 min. After the urea was reduced to 1 M by 50 mM ammonium bicarbonate (pH 8.0), the protein samples were finally digested with trypsin at an enzyme:protein mass ratio of 1:50 for 24 h at 37 °C [28].

### 3.4. Lipidomic Analysis Using LC-MS

The separation of lipids was achieved at 55 °C using a BEH C18 column (2.1 mm × 100 mm, 1.7 µm) on Waters ACQUITY UPLC-Xevo G2 Q-TOF (Waters Corporation, Milford, USA). The mobile phase A and B were acetonitrile/water (ACN/H_2_O, 60:40, *v*/*v*), and isopropanol/acetonitrile (IPA/CAN, 90:10, *v*/*v*) respectively, both containing 0.1% formic acid and 5 mM ammonium [29]. The gradient program used for chromatographic separation was as follows: 0–2 min, 40–43% B; 2–2.1 min, 43–50% B; 2.1–12 min, 50–54% B; 12–12.1 min, 54–70% B; 12.1–18 min, 70–99% B; 18–18.1 min, 99–40% B; 18.1–20 min, 40% B. The flow rate was set to 0.3 mL/min. The injection volume was 10 μL.

The mass spectra were acquired using electrospray ionization in positive mode. The settings used for the MS analysis were as follows: capillary voltage: 2500 V, sampling cone voltage: 35 V, desolvation temperature: 350 °C, desolvation gas flow: 700 L/h, cone gas flow: 50 L/h, source temperature: 400 °C. The mass (*m*/*z*) range was from 50 to 1200. The Masslynx 4.1 software (Waters Corporation, Milford, USA) was used for data acquisition and MZmine was employed to remove the noise from the total ion chromatogram and to extract the mass spectrum peaks. After merging and aligning, the three-dimensional matrix dataset was obtained. The obtained data were imported into the SIMCA-P13.0 software (Umetrics, Umeå, Sweden) for PCA and OPLS-DA. The differential lipids were identified based on the accurate MS data and HMDB database.

### 3.5. Proteomic Analysis Using Nanorplc-MS/MS.

The peptides were dissolved in a 0.1% solution of formic acid (FA) in water, and then analyzed on 2-D nanoLC (Eksigent, Silicon Valley, California, USA) coupled to 5600 LTQ-Orbitrap Velos (Thermo Fisher, 81 Wyman Street, Waltham, MA, USA) using a C18 column (3.6 mm × 100 mm, 3 μm), at a solvent flow rate of 350 nL/min. The mobile phase consisted of 0.1% FA in water (solvent A) and acetonitrile (solvent B). The samples were eluted using the following linear gradient: 5–8% B, 5 min; 8–18% B, 35 min; 18–32% B, 22 min; 32–95% B, 2 min; 95% B, 4 min, 95–5% B, 4 min.

For MS analysis, a nano-spray ion source was employed. A spray voltage of 1800 V was applied and the ion transfer tube was at 350 °C. The mass spectrometer was programmed to acquire in a data-dependent mode. The MS scan range was from an *m*/*z* of 375 to an *m*/*z* of 1600, with a resolution of 60,000 at an *m*/*z* of 400. The 50 most intense peaks with charge state 2 and above were acquired by collision-induced dissociation with a normalized collision energy of 35% and activation time of 5 ms, using one microscan, and the intensity threshold was set at 500. The MS^2^ spectra were acquired in the LTQ normal scan mode. The proteins were identified by Proteome Discoverer version 1.3 using MASCOT search engine with percolator against the mouse RefSeq protein database (updated on 17 November 2016). The mass tolerance was set to 20 ppm for the precursor. For the tolerance of product ions, Velos was set as 0.5 Da. Oxidation (M) were chosen as variable modifications; carbamidomethylation (C) as fixed modification, two missed cleavage sites for trypsin was allowed. The false discovery rates (FDRs) for both the peptides and proteins were controlled to be lower than 1%.

The intensity-based absolute quantification (iBAQ) of proteins was performed using an in-house developed software. The differentially regulated proteins (with P values less than 0.01) were screened using ANOVA with Mev.4.9.0 software (Oracle Corporation, Redwood City, California, USA). The functional annotation of the proteins and identification of the biological processes in which they were involved was done using Uniprot, GO, and KEGG databases.

## 4. Results and Discussion

### 4.1. Neuroprotective Effect of XA on MPP ^+^-Induced MN9D Cells

#### Optimization of Suitable Concentrations of MPP^+^ and XA

The MPP^+^-injured MN9D cell model was established in the present study. The results of CCK-8 test demonstrated a dose-dependent relationship between cell injury and MPP^+^ concentration (Table 2). According to CCK-8 assay, the survival rate of 500 μM MPP+ was 50.44 ± 1.05 (%). Thus, the IC50 value of MPP+ was 500 μM after 24 h incubation. 500 μM MPP^+^ was the suitable concentration for 24 h on MN9D cells.

The results of preliminary experiments revealed that 100 μM XA could significantly inhibit MPP^+^-induced neuron loss in MN9D cells (Table 3). The appropriate concentration range of XA was determined to be 50–100 μM. These findings suggested that XA possessed a neuroprotective effect on MPTP-induced dopaminergic neurons in MN9D cells.

The cell morphology was evaluated by visualization under an inverted microscope. From the cell microscope, the changes in cell morphology could be also observed. The control group of MN9D cells mostly adhered to the wall and were enated; most also had protrusions. After the MN9D cells had been treated with 500 μM MPP^+^ for 24 h, the number of adherent cells decreased and protuberances disappeared. Thus, 500 μM was chosen as the model concentration. When the concentration of XA was 100 μM, the number of adherent cells was increased, and most of the cells had protrusive growth recovery, and gradually extended outward under the inverted microscope (Figure 2).

### 4.2. Absolute and Relative Quantification of Neurotransmitters

#### Method Validation

The standard solutions containing the internal standard were diluted with 50% methanol to six different concentrations for construction of calibration curves. The ratio of peak area of each neurotransmitter to that of the internal standard (Yi/Ys) was plotted against the concentration (X, ng/mL). All the calibration curves indicated good linearity with correlation coefficients (r) ranging from 0.9960 to 0.9999. The limits of detection (LOD; S/N = 3/1) and the limits of quantification (LOQ; S/N = 10/1) ranged from ~0.954 to 5.711 ng/mL and from ~2.386 to 11.423 ng/mL (Table 4).

The precision of the present method was calculated by analyzing the standard solution under the optimized experimental conditions. The RSDs (%) were ~0.45–3.10 (*n* = 6). Furthermore, the sample solutions were prepared in parallel (*n* = 5) to evaluate the repeatability and RSDs (%) of ~2.02–5.91 were achieved. The RSDs (%) for the stability of each constituent after 48 h at room temperature (*n* = 6) were ~1.51–6.97 (Appendix A).

### 4.3. Quantification of Neurotransmitters

The concentrations of 27 neurotransmitters in MN9D cells and the cell supernatant, before and after the treatment with XA, were measured using UPLC-QqQ MS/MS. Absolute quantification was achieved for 10 neurotransmitters, whereas relative quantification was done for 17 neurotransmitters (Table 5 and Table 6 and Appendix A). The concentration of taurine, norepinephrine, choline, acetylcholine, butyrylcholine, homovanillic acid, proline, histamine, serotonin, adrenaline, and levodopa displayed the same trend of variation in a dose-dependent manner in both the cell supernatant and MN9D cells. However, XA showed little adjustment for the rest of the neurotransmitters. The concentration of these neurotransmitters decreased in the model group, whereas they increased after the administration of XA for 48 h. Taurine, norepinephrine, and cholinergic neurotransmitters were the dominant neurotransmitters. Taurine participates in a number of neuroprotective processes, which directly affect neurodevelopment and neuronal excitability. The mechanism for the observed effect of taurine was reported to be closely related to the prevention of mitochondrial dysfunction and to the protection against endoplasmic reticulum (ER) stress [30,31]. Also, norepinephrine could protect cultured neurons from oxidative stress and amyloid-induced toxicity [32]. Cholinergic neurotransmitters were reported to be a significant neural medium for maintaining advanced neural activities [33,34]. The levels of acetylcholine and butyrylcholine determined in this study were in accordance with those obtained in our previous study [7]. 

### 4.4. XA Regulates Distinct Lipid Profiles in MPP^+^-Induced MN9D Cell Model

We used UPLC-Q-TOF technology to study the lipid profile of MPP+-induced MN9D cells after XA treatment. The obtained data were analyzed using PCA and OPLS-DA (Figure 3 and Figure 4). PCA results indicated that the samples could gather together in control, model, and administration groups. OPLS-DA score chart results demonstrated that these 3 groups were significantly separated. These parameters can also indicate that the establishment of MPP^+^-injured MN9D cells model is successful, which has a greater impact on the lipid metabolism network of MN9D cells, and the lipid metabolism disorder will be adjusted after administration of XA. The results from S-plot loadings plots showed the differentially-expressed lipids between two groups, namely the control and model groups, and the model and administration groups. The RSD values for each group were less than 30%. Variable importance in the project (VIP) was an important parameter of differential metabolites. The metabolite concentration with VIP > 1.5 and *p* < 0.05, as determined by t test, were considered to be significantly different. The metabolic profile of the XA group differed greatly from that of the MPP+ group, and was close to that of the control group. We observed that MPP+ injury mainly influenced the metabolism of PC, LysoPC, ceramide (Cer), and PE. In all, 22 differentially-expressed lipids were identified after MPP+ injury and XA intervention, including 10 PCs, two LysoPCs, two PEs, four Cers, one DG, and three others (Table 7). PC and PE are the most abundant phospholipids in all the mammalian cell membranes, which are important for the neuroprotective effect in neurodegenerative diseases as potential neuroprotective agents [35,36]. Take PC (0:0/18:0) as an example, breviscapine ameliorated the learning and memory deficits of AD mice predominantly by regulating phospholipids metabolism. Among of them, PC (0:0/18:0) was one of potential biomarkers [37]. LysoPCs are important inter- and intra- cellular lipid mediators, which also regulate the nervous system. A LysoPC + FA mixture was reported to increase both the spontaneous and evoked release of neurotransmitters [38,39]. As for LysoPC(O-18:0), exposure to acephate disrupted metabolism of lipids, including lysoPC (15:0), lysoPC (16:0), lysoPC (O-18:0), lysoPC (18:1(9Z)), lysoPC (18:0), lysoPC (20: 4(5Z, 8Z,11Z,14Z)), which induced oxidative stress and caused neurotoxicity [40]. Cer, the precursor of all complex sphingolipids, is an important bioactive lipid with roles in several biological processes, and has been reported to be involved in age-related, neurological, and neuroinflammatory diseases [41,42]. As reported in the literature [43], C16- and C18-ceramides were closely associated with neurodegeration. Notably, the administration of XA effectively regulated differential lipid metabolism.

### 4.5. XA Regulates Distinct Protein Profiles in MPP^+^-induced MN9D Cell Model

The proteomic profile of MPP^+^-injured MN9D cells after XA treatment was analyzed by LTQ-Orbitrap MS/MS. A total of 4299 proteins were detected. Among the proteins that were differentially-regulated in a dose-dependent manner after XA treatment, 142 were upregulated and 209 were downregulated (Appendix A). Moreover, we identified 60 common differential proteins in different groups using ANOVA, performed with the Mev.4.9.0 software (*p* < 0.01). Functional analysis revealed that these proteins were mainly related to the central nervous system projection, neuron axonogenesis, phosphatidylinositol-mediated signaling, lipoprotein transport, and lipid metabolism. Among these, 34 proteins were highly-abundant and in correlation with neuroprotective effects according to functional analysis. The heat map of 34 differentially-expressed proteins is presented in Figure 5. Considering the network of proteins and results reported in previous studies, 11 differentially-expressed proteins displayed biological significance with regard to the neuroprotective effect (Table 8). The expression of Fructose-bisphosphate aldolase C Aldoc, Dihydrofolate reductase (Dhfr), ATP synthase subunit delta, mitochondrial (Atp5d), Lysosome membrane protein 2 (Scarb2), Adenylate kinase isoenzyme 1(Ak1), and Vesicle transport through interaction with t-SNAREs homolog 1B (Vti1b) was decreased in the MPP+-induced cell model, whereas that of Casein kinase II subunit alpha’(Csnk2a2), Heme oxygenase 1(Hmox1), BCL2/adenovirus E1B 19 kDa protein-interacting protein 2(Bnip2), 39S ribosomal protein L50, mitochondrial (Mrpl50), and Alpha-galactosidase A (Gla) was increased. Aldoc is an isozyme of aldolase that exists in the brain and nervous system, and is reported to directly affect S-nitrosylation in AD regions [44,45]. The deficiency of Dhfr led to severe neurological diseases [46]. Atp5d, which is related to ATP metabolism, was down regulated in AD samples [47]. Scarb2 was reported to be associated with PD [48]. Ak1 decreased significantly upon neurological damage [49]. The lack of Vti1b led to impairments in the neuronal development [50]. Nevertheless, XA could restore the expression levels of the aforementioned proteins to that in the control group, which was markedly elevated by ~1.62–3.04 times compared to the level in the model group. Emerging evidence suggests that these proteins are mainly involved in energy metabolism. Furthermore, in leukocytes, the expression of Csnk2a2 was related to the telomere length, which is directly associated with age-related diseases [51,52]. Increased expression of Hmox1 was detected in the brain of APP swe/PS1 transgenic mice [53]. The increase in Bnip2 expression displayed strong correlation with the apoptosis induced by neurotoxic agents [54]. High expression of Mrpl50 was found to correlate with mitochondrial translation using GO annotation analysis. The expression of Gla inhibited neuronal growth [55]. Intriguingly, the XA group exhibited low levels of Csnk2a2, Hmox1, Bnip2, Mrpl50, and Gla, with a ~0.34–0.69 fold decrease compared to the expression of the respective genes in the MPP^+^-treated group.

### 4.6. Potential Pathways Involved in the Neuroprotective Effect of XA

We established a network of differentially-expressed lipids and proteins based on HMDB and KEGG databases (Figure 6). The relationship of each node protein was clarified and built directly or indirectly in the network. Furthermore, the differentially-expressed proteins were correlated with some differentially-expressed lipids based on the protein–lipid interaction determined using HMDB. Csnk2a2 was observed to potentially interact with PC (O-18:1(9Z)/2:0) and LysoPC (O-18:0). Scarb2, Gla, and Vti1b prominently regulated Cers, namely LacCer (d18:1/24:0), Trihexosylceramide (d18:1/24:0), LacCer (d18:1/22:0), and LacCer (d18:1/24:1(15Z). Coenzyme Q8 was determined to possibly participate in the regulation of energy metabolism by Atp5d and Ak1. Mrpl50 was observed to be the regulatory protein for many PCs and PEs, including PC (O-18:1(9Z)/2:0), PC (20:0/22:4(7Z,10Z,13Z,16Z)), PC (P-16:0/22:4(7Z,10Z,13Z,16Z)), PC (16:1(9Z)/22:4(7Z,10Z,13Z,16Z), PE (20:0/22:4(7Z,10Z,13Z,16Z)), and PE (P-16:0/0:0). We also determined the possible pathways in which these node proteins were involved. Gla is a key protein involved in lysosome, sphingolipid metabolism, and glycosphingolipid biosynthesis (globo series). In a previous study [9], it was observed that MPP^+^ could enhance sphingolipid metabolism. Moreover, Atp5d, Dhfr, Ak1, and Aldoc participate in energy metabolism and in the regulation of mitochondrial function. Mounting evidence indicates that mitochondrial dysfunction is involved in the aging of brain and in neurodegenerative diseases. Seven differentially-expressed lipids, including PC (0:0/18:0), PC (24:0/0:0), PC (P-18:0/0:0), PC (16:1(9Z)/22:4(7Z,10Z,13Z,16Z)), LysoPC (0-18:0), PE (P-16:0/0:0), and PC (18:1(9Z)/22:4(7Z,10Z,13Z,16Z)), were validated by their relative peak areas using UPLC-Orbitrap MS/MS. The validation data were in accordance with the results of the lipidomic analysis (Appendix A). To verify the accuracy of the proteomics method, four proteins (Scarb2, Csnk2a2, Vti1b, and Bnip2) were validated by an ELISA test according to the instructions in each kit, and similar results were obtained (Appendix A). In summary, the overall results suggest that the neuroprotective effects of *N*-benzylhexadecanamide are mainly through the regulation of sphingolipid metabolism and mitochondrial function.

## 5. Conclusions

In the present investigation, a method for the simultaneous determination of 27 intracellular and extracellular neurotransmitters was established using MN9D cells. The HR-MS/MS technology was used comprehensively for lipidomics and proteomics to decipher the mechanism underlying the neuroprotective effect of XA. The major differentially-expressed lipids and proteins were validated by semiquantitative ELISA methods. The results indicated that XA could regulate the monoamine neurotransmitters, PCs, PEs, and Cers, and node proteins related to energy metabolism. All these differential biomarkers support that the neuroprotective mechanism of XA which is mainly involved in the regulation of sphingolipid metabolism and mitochondrial function. Additionally, the network of proteins and lipids also supported this conclusion. This study has revealed a possible mechanism underlying the neuroprotective effect of XA, which should provide clues for further investigation of XA in vivo. MPP+ is reportedly mediated through oxidative mechanisms by inhibiting NADH-CoQ10 reductase (complex I) of the respiratory chain in mitochondria, and then generating reactive oxygen species (ROS) [56,57]. It is a well-established fact that oxidative stress mechanisms become more prominent with aging and the development of neurodegenerative disorders [58,59]. Our results demonstrated for the first time that the neuroprotective effects of XA were accompanied by an improvement of mitochondrial respiratory function. Nevertheless, further studies are required to clarify the cause-and-effect relationship between the reduction of oxidative stress and the improvement of mitochondrial function by XA.

## Figures and Tables

**Figure 1 molecules-23-02929-f001:**
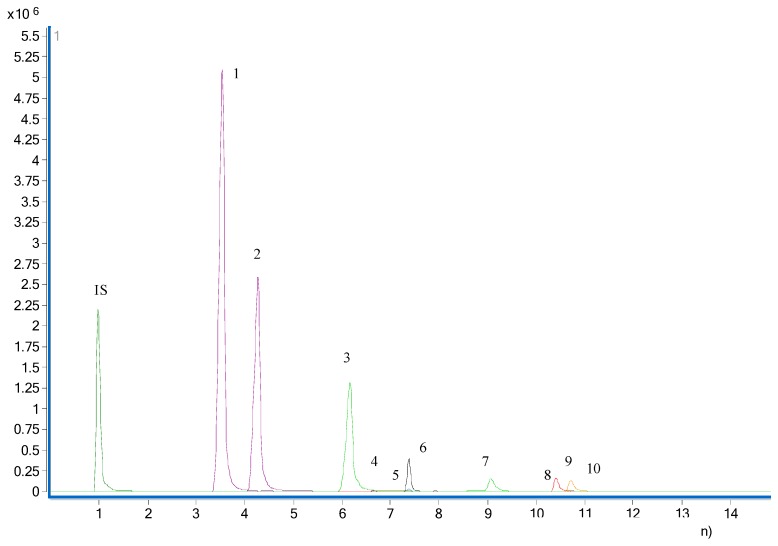
MRM chromatogram of the 10 neurotransmitters. 1, butyrylcholine; 2, acetylcholine; 3, choline; 4, taurine; 5, homovanillic acid; 6, tyrosine; 7, noradrenaline; 8, serine; 9, aspartic acid; 10, asparagine.

**Figure 2 molecules-23-02929-f002:**
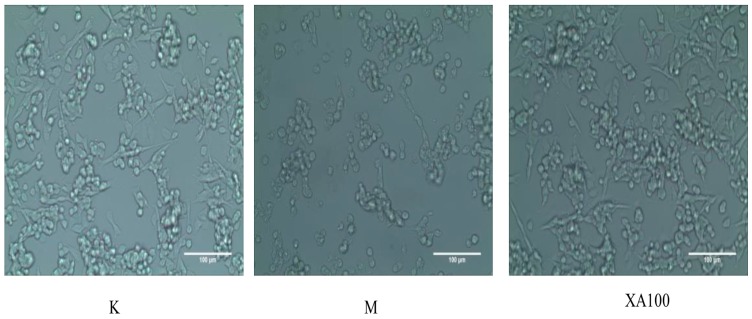
Morphological changes in MN9D cells as observed under an inverted microscope (×200). K: Control group; M: Model group; XA100: 100 μM XA.

**Figure 3 molecules-23-02929-f003:**
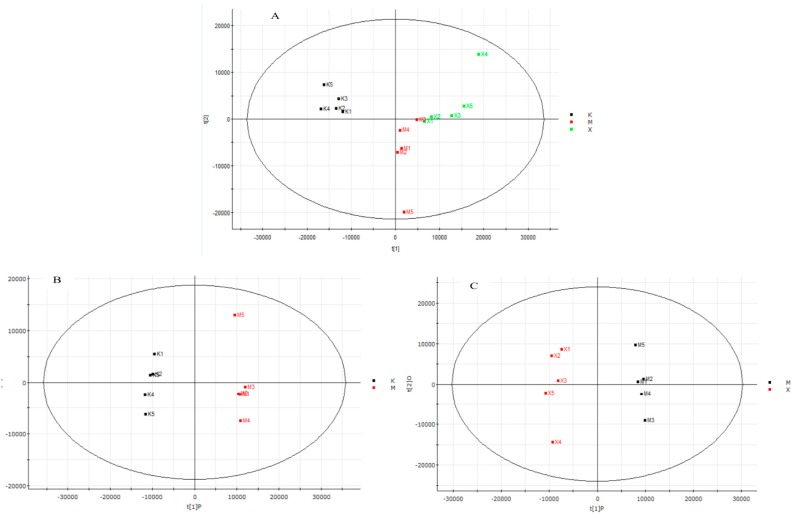
PCA and OPLS-DA score charts of control (K), model (M), administration (X) group. (**A**). PCA score chart of K, M and X group; (**B**): OPLS-DA score charts of K and M group; (**C**). OPLS-DA score charts of M and X group.

**Figure 4 molecules-23-02929-f004:**
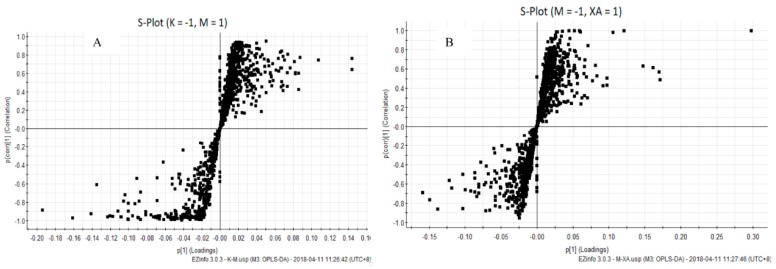
S-plot loadings plots of Control vs. Model group and Model vs. Administration group. (**A**): Control vs. Model group; (**B**): Model vs. Administration group.

**Figure 5 molecules-23-02929-f005:**
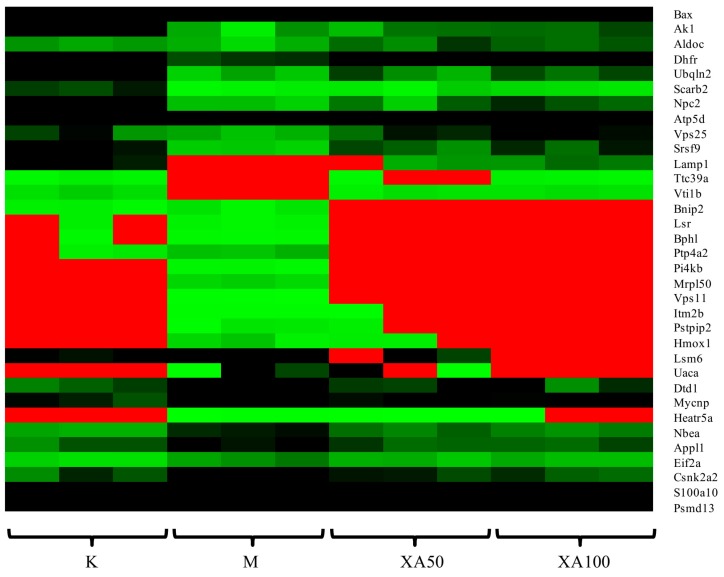
Heat map analysis of 34 differentially expressed proteins. K: Control group; M: Model group; XA50: 50 μM XA; XA100: 100 μM XA.

**Figure 6 molecules-23-02929-f006:**
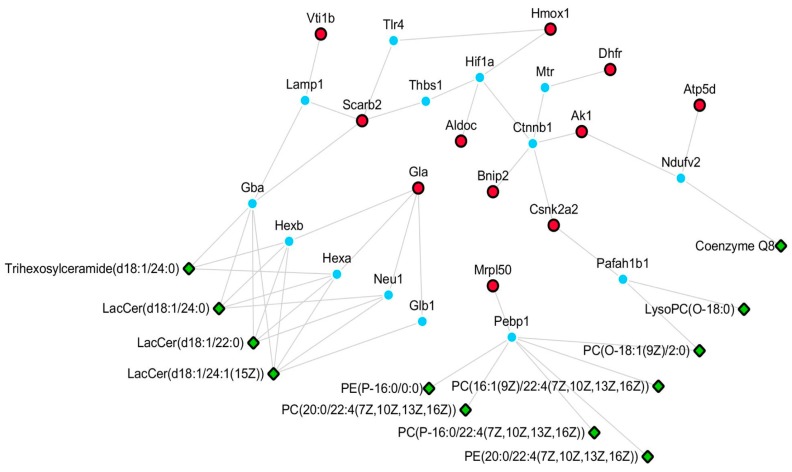
Network of differentially-expressed proteins and lipids. Green part: differentially-expressed lipids of lipidomics study; Red part: differentially-expressed proteins of proteomics; Blue part: upstream proteins of metabolites.

**Table 1 molecules-23-02929-t001:** Optimized mass transition ion pairs (*m*/*z*) and CE values for neurotransmitters.

No.	Chemical Compounds	Quant/Qual Ion Pair	Quant/Qual CE(V)
0	diazepam	285→193/285→154	37/30
1	aspartic acid	134→88/134→116	6/6
2	asparagine	133→87.1/133→74.1	6/16
3	serine	106→60.1/106→88.1	9/6
4	taurine	126→108/126→44.2	9/19
5	tyrosine	182→165.0/182→136.0	16/12
6	noradrenaline	170→152.0/170→134.0	6/18
7	homovanillic acid	183→137.0/183→137.0	14/14
8	choline	104→59.9/104→45.1	18/25
9	acetyl choline	146→87.0/146→60.2	10/6
10	butyrylcholine	174→115.0/174→71	16/20
11	ornithine	133→70/133→116	21/6
12	lysine	130→84/130→56	15/37
13	phenylalanine	166→120/166→103	15/25
14	tryptophan	205→188/205→146	10/15
15	L-leucine	132→86/132→30	9/12
16	methionine	150→104/150→56	12/9
17	dimethylglycine	104→58/104→42	15/37
18	proline	116→69.9/116→69.9	3/3
19	histamine	112→94.9/112→67.9	12/21
20	threonine	120→74/120→101.9	6/6
21	citrulline	176→159/176→70	6/24
22	arginine	175→70/175→116	18/15
23	serotonin	177.1→160/177.1→115.0	10/28
24	adrenaline	184→166.0/184→135.1	6/16
25	dopamine	154→137.0/154→91	10/29
26	levodopa	198→181.0/198→152.0	6/19
27	γ-aminobutyric acid	104→87.1/104→69.1	9/16

**Table 2 molecules-23-02929-t002:** Survival rate of MPP^+^-injured MN9D cells.

Treatment	Survival Rate (%)
Control	100
1000 μM MPP^+^	41.34 ± 1.16
500 μM MPP^+^	50.44 ± 1.05
250 μM MPP^+^	72.16 ± 2.36
125 μM MPP^+^	79.62 ± 2.25
67.5 μM MPP^+^	80.49 ± 2.71

**Table 3 molecules-23-02929-t003:** The survival rate of XA on MPP^+^-injured on MN9D cells after 48 h treatment (*n* = 3).

Concentration	Survival Rate (%)
Control group	100
Model group	49.94 ± 0.26
200 μM XA	62.07 ± 1.78 *
100 μM XA	79.27 ± 2.23 **
50 μM XA	54.56 ± 1.35
25 μM XA	50.75 ± 2.01
12.5 μM XA	54.91 ± 2.28

Note: The concentration of DMSO was less than 0.1%; the model group was 500 μM MPP^+^; * *p* < 0.05, ** *p* < 0.01.

**Table 4 molecules-23-02929-t004:** Regression equations, correlation coefficient, detection limits, and linear range of 10 neurotransmitters.

Analyte	Regression Equation	R^2^	LOD (ng/mL)	LOQ (ng/mL)	Linear Range (ng/mL)
aspartic acid	Y = (0109 × X) + 2.431 × 10^−4^	0.9961	2.510	10.040	10.040–2008
asparagine	Y = (1.087 × X) + 3.066 × 10^−4^	0.9999	1.925	4.813	4.813–3850
serine	Y = (2.239 × X) + 6.326 × 10^−3^	0.9971	2.436	4.871	4.871–1948.5
taurine	Y = (0.137 × X) + 2.844 × 10^−4^	0.9960	2.471	4.941	4.941–1976.5
tyrosine	Y = (1.880 × X) + 2.142 × 10^−3^	0.9988	1.988	4.971	4.971–3976.5
noradrenaline	Y = (2.765 × X) − 7.771 × 10^−5^	0.9995	1.065	2.663	2.663–2130.5
homovanillic acid	Y = (0.610 × X) + 3.685 × 10^−4^	0.9986	2.471	2.471	2.471–1976.5
choline	Y = (10.050 × X) + 3.276 × 10^−2^	0.9972	5.711	11.423	11.423–4569
acetylcholine	Y = (39.80 × X) + 0.040	0.9972	1.018	2.544	2.544–2035.5
butyrylcholine	Y = (74.40 × X) + 0.045	0.9993	0.954	2.386	2.386–1909

**Table 5 molecules-23-02929-t005:** Results of absolute quantification of neurotransmitters in cell supernatants and cells (ng).

Analyte	S				C			
	K	M	XA50	XA100	K	M	XA50	XA100
aspartic acid	2395.355 ± 96.529	5510.444 ± 467.377 **	14274.770 ± 1087.576 **##	23712.580 ± 4528.323 *#	1944.144 ± 109.031	1375.344 ± 83.746 **	3700.921 ± 185.859 **##	4927.835 ± 288.260 **##
asparagine	6884.435 ± 1015.190	14,272.680 ± 1663.724 *	31,556.410 ± 3124.580 **##	40,738.920 ± 2586.551 **##	10,116.610 ± 1352.500	4504.988± 290.293 *	4213.556 ± 73.306 *	3824.943 ± 190.385 *#
serine	13,975.260 ± 1087.511	39,178.200 ± 5911.520 *	60,529.650 ± 3594.469 **#	80,162.380 ± 1577.983 **##	12,317.940 ± 1561.040	6873.225 ± 1076.321 **	5501.029 ± 18.212 *	5555.612 ± 229.131 *
taurine	471.331 ± 99.991	142.098 ± 10.374 *	432.399 ± 71.048 #	1105.402 ± 156.281 *##	20,115.850 ± 2635.606	9444.964 ± 807.414 *	12,097.300 ± 261.632 *#	17,170.847 ± 831.887 *##
tyrosine	236.723 ± 8.283	487.634 ± 6.951 **	591.228 ± 29.906 **#	707.278 ± 67.708 **#	708.870 ± 81.875	294.214 ± 34.469 **	509.265 ± 33.050 #	395.586 ± 29.330 *
noradrenaline	118.122 ± 6.944	113.990 ± 4.223	125.690 ± 4.836	136.567 ± 3.149 #	90.074 ± 6.222	66.727 ± 2.370 *	68.401 ± 1.072 *	69.707 ± 1.770 *#
homovanillic acid	320.472 ± 9.119	235.988 ± 10.898 **	317.552 ± 20.965 **#	396.112 ± 31.864 **#	480.939 ± 84.673	205.297 ± 22.389 *	271.848 ± 15.054 #	461.287 ± 35.156 #
choline	115.893 ± 21.214	61.222 ± 7.910 *	254.697 ± 5.314 *##	266.724 ± 9.861 **##	6.203 ± 1.699	5.953 ± 0.353 *	-	-
acetylcholine	423.041 ± 64.848	227.654 ± 101.324 *	630.147 ± 44.006 ##	1108.037± 101.248 **##	1600.918 ± 143.872	380.128 ± 31.322 **	440.118 ± 10.453 **	582.330 ± 5.916 **#
butyrylcholine	-	-	4.309 ± 0.752 **##	7.731 ± 0.434 **##	9.904 ± 0.761	8.152 ± 0.566	10.206 ± 0.807	28.001 ± 2.426 **##

* *p* < 0.05, ** *p* < 0.01 (vs. control group), # *p* < 0.05, ## *p* < 0.01 (vs. model group), S: cell supernatant, C: cells, K: control group, M: model group, XA50: 50 μM XA, XA100: 100 μM XA.

**Table 6 molecules-23-02929-t006:** Results of relative quantification of neurotransmitters in cell supernatants and cells ((A_x_/A_i_) × 100).

Analyte	S				C			
	K	M	XA50	XA100	K	M	XA50	XA100
ornithine	600.781 ± 100.501	1517.943 ± 117.101 **	2062.528 ± 122.675 **##	1548.961 ± 48.987 **	32.226 ± 3.387	21.717 ± 0.813 **	19.457 ± 0.142 **##	18.397 ± 1.329 **#
lysine	27.421 ± 4.076	14.576 ± 0.558 **	11.510 ± 0.050 **#	11.346 ± 0.043 **##	0.051 ± 2.57 × 10^−4^	0.041 ± 5.78 × 10^−3^ *	0.050 ± 7.93 × 10^−3^	0.043 ± 6.83 × 10^−3^
phenylalanine	21.522 ± 4.642	90.387 ± 6.830 **	130.374 ± 12.928 **##	111.406 ± 7.036 **##	7.909 ± 2.583	5.551 ± 0.305	4.707 ± 0.193 #	4.252 ± 0.259 ##
tryptophan	0.474 ± 6.62×10^−2^	0.258 ± 7.24 × 10^−3^ **	0.289 ± 4.87 × 10^−2^ *	0.198 ± 2.97 × 10^−3^ **##	6.70 × 10^−2^ ± 6.25 × 10^−3^	3.30×10^−2^ ± 2.33 × 10^−3^ **	2.65×10^−2^ ± 1.81 × 10^−3^ **#	3.26×10^−2^ ± 2.17 × 10^−3^ **
L-leucine	63.876 ± 10.329	733.788 ± 219.994 **	1042.663 ± 180.334 **	857.797 ± 37.087 **	94.712 ± 4.736	86.116 ± 1.129 *	79.111 ± 1.922 **##	70.187 ± 8.315 *#
methionine	0.784 ± 0.129	3.676 ± 0.805 **	7.124 ± 0.925 **##	5.232 ± 0.601 **	0.388 ± 0.048	0.289 ± 0.016 *	0.300 ± 0.013 *	0.246 ± 0.028 *
dimethylglycine	4.397 ± 0.655	9.808 ± 1.845 **	12.696 ± 1.353 **	12.398 ± 0.745 **	1.505 ± 0.003	1.400 ± 0.091	1.497 ± 0.057	1.490 ± 0.086
proline	34.412 ± 2.978	26.567 ± 3.802	55.397 ± 6.297	62.614 ± 2.243	17.092 ± 2.391	5.750 ± 0.444	6.133 ± 0.622	8.218 ± 0.561
histamine	0.192 ± 2.94 × 10^−2^	0.174 ± 3.80 ×10^−3^	0.256 ± 1.42 × 10^−2^*##	0.281 ± 1.73×10^−2^ #	1.91×10^−2^ ± 7.11 × 10^−3^	0.92×10^−2^± 1.14 × 10^−3^	1.09×10^−2^ ± 1.75 × 10^−3^	1.09×10^−2^ ± 5.62 × 10^−4^
threonine	13.792 ± 1.807	42.426 ± 0.420 **	58.133 ± 2.274 **##	42.037 ± 4.344 **	7.283 ± 2.135	3.710 ± 0.640 *	3.766 ± 8.74×10^−2^ *	4.032 ± 0.281
citrulline	52.390 ± 4.242	163.451 ± 23.996 **	238.912 ± 31.804 **#	224.588 ± 20.786 **#	6.605 ± 1.205	2.862 ± 0.056 **	3.027 ± 0.085 **#	2.853 ± 0.165 **
arginine	6.172 ± 0.028	8.938 ± 0.888 **	12.233 ± 0.267 **	11.151 ± 0.956 **#	0.380 ± 0.060	0.888 ± 0.021 *	0.267 ± 0.021 *	0.252 ± 0.03 *
serotonin	4.91 × 10^−3^ ± 7.27 × 10^−4^	7.14 × 10^−4^ ± 8.65 × 10^−5^ **	1.20 ×10^−3^± 2.67 × 10^−4^*	4.89×10^−3^ ± 4.33 × 10^−4^ ##	1.25×10^−3^ ± 3.16 × 10^−4^	9.70×10^−4^ ± 7.74 × 10^−5^	1.15×10^−3^ ± 1.61 × 10^−5^	1.29×10^−3^ ± 1.01 × 10^−4^
adrenaline	4.57 × 10^−3^ ± 7.95e × 10^−4^	1.24 × 10^−3^ ± 1.43e × 10^−4^ *	1.36×10^−3^ ± 1.02 × 10^−4^ *	1.82×10^−3^ ± 9.45 × 10^−5^ *##	8.65×10^−4^ ± 8.35 × 10^−5^	4.37×10^−4^ ± 5.54 × 10^−5^**	4.90×10^−4^ ± 5.25 × 10^−5^ **	1.25×10^−3^ ± 1.76×10^−4^ *##
dopamine	2.40 × 10^−3^ ± 2.68e × 10^−4^	2.07 × 10^−3^ ± 7.92 × 10^−5^	2.68×10^−3^ ± 1.67 × 10^−4^#	1.44×10^−3^ ± 9.89 × 10^−5^ *#	3.41×10^−3^ ± 7.10 × 10^−4^	1.71×10^−4^ ± 1.71 × 10^−4^ *	2.70×10^−3^ ± 6.64 × 10^−4^	1.10×10^−3^ ± 7.37 × 10^−5^ *
levodopa	1.41 × 10^−3^ ± 1.42 × 10^−4^	1.17 × 10^−3^ ± 2.85 × 10^−4^	1.77×10^−3^ ± 1.97 × 10^−4^	1.84×10^−3^ ± 1.69 × 10^−4^	1.63×10^−3^ ± 2.31 × 10^−4^	5.80×10^−4^± 8.57 × 10^−5^ *	1.14×10^−3^ ± 8.99 × 10^−5^ ##	1.43×10^−3^ ± 6.93×10^−5^ *##
γ-aminobutyric acid	2.1 × 10^−2^ ± 4.22 × 10^−3^	2.83 × 10^−3^ ± 1.70 × 10^−3^ *	3.30×10^−3^ ± 5.53 × 10^−4^ *	3.23×10^−3^ ± 1.52 × 10^−4^ *	7.47×10^−4^ ± 1.52 × 10^−4^	6.14×10^−4^ ± 1.90 × 10^−5^	7.81×10^−4^ ± 1.03 × 10^−4^	7.40×10^−4^ ± 8.50 × 10^−4^

* *p* < 0.05, ** *p* < 0.01 (vs. control group), # *p* < 0.05, ## *p* < 0.01(vs. model group), S: cell supernatant, C: cells, K: control group, M: model group, XA50: 50 μM XA, XA100: 100 μM XA (Ax/Ai) × 100: (the peak area of the neurotransmitter/the peak area of the internal standard) × 100.

**Table 7 molecules-23-02929-t007:** Differentially expressed lipids in the control, model, and administration groups.

No.	Potential Biomarkers	t_R_/Min	*m*/*z*	Formula	VIP		Type	Change Multiplier (M/K)	Change Multiplier (XA/M)	Post-Model Trend
					K-M	XA-M				
L1	PC(O-18:1(9Z)/2:0)	2.82	550.3870	C32H64NO7P	1.31	1.57	PC	0.46	2.07	↓
L2	PC (0:0/18:0)	2.70	524.3714	C26H54NO7P	4.76	4.47	PC	0.48	1.67	↓
L3	PC (24:0/0:0)	6.59	608.4654	C32H66NO7P	2.29	1.74	PC	0.50	1.46	↓
L4	PC(P-18:0/0:0)	3.08	508.3761	C26H54NO6P	3.47	1.27	PC	0.79	1.87	↓
L5	PC (18:2(9Z,12Z)/22:2(13Z,16Z))	12.39	860.6162	C48H88NO8P	1.73	2.05	PC	1.57	0.66	↑
L6	PC (P-16:0/22:4(7Z,10Z,13Z,16Z))	10.64	816.5897	C46H84NO7P	2.17	2.09	PC	2.13	0.63	↑
L7	PC (18:3(6Z,9Z,12Z)/22:2(13Z,16Z))	10.02	858.6007	C48H86NO8P	1.01	1.14	PC	2.77	0.42	↑
L8	PC (16:1(9Z)/22:4(7Z,10Z,13Z,16Z))	7.50	830.5687	C46H82NO8P	1.59	1.76	PC	1.57	0.67	↑
L9	PC (18:1(9Z)/22:4(7Z,10Z,13Z,16Z))	10.44	858.6012	C46H86NO8P	1.93	1.37	PC	1.84	0.38	↑
L10	PC (20:0/22:4(7Z,10Z,13Z,16Z))	13.33	888.6476	C50H92NO8P	2.28	1.20	PC	3.82	0.70	↑
L11	LysoPC (24:1(15Z))	4.87	606.4498	C32H64NO7P	1.53	1.18	LysoPC	0.46	2.70	↓
L12	LysoPC (O-18:0)	3.63	552.4031	C26H56NO6P	1.40	2.03	LysoPC	0.50	1.70	↓
L13	PE (P-16:0/0:0)	2.31	438.2983	C21H44NO6P	1.10	1.04	PE	0.41	1.86	↓
L14	PE (20:0/22:4(7Z,10Z,13Z,16Z))	13.21	824.6181	C47H86NO8P	2.60	2.21	PE	1.67	0.71	↑
L15	LacCer (d18:1/24:0)	14.70	974.7503	C54H103NO13	1.54	3.79	Cer	0.62	3.23	↓
L16	Trihexosylceramide (d18:1/24:0)	14.51	1136.8030	C60H113NO18	2.11	1.80	Cer	0.59	3.99	↓
L17	LacCer (d18:1/22:0)	14.16	946.7188	C52H99NO13	2.00	1.64	Cer	0.45	3.62	↓
L18	LacCer (d18:1/24:1(15Z))	14.14	972.7347	C54H101NO13	1.40	1.72	Cer	0.50	3.98	↓
L19	DG (16:0/18:1(9Z)/0:0)	13.07	603.5351	C37H70O5	1.58	1.55	DG	1.82	0.67	↑
L20	Coenzyme Q8	15.33	727.5657	C49H74O4	1.31	1.08	Other	1.46	0.69	↑
L21	Montecristin	10.41	575.5036	C37H66O4	1.43	1.51	Other	1.62	0.68	↑
L22	Calcitriol	18.31	369.3520	C27H44O3	3.76	2.44	Other	0.44	1.45	↓

K: control group, M: model group, XA: 100 μM XA. PC: phosphatidylcholine; PE: phosphatidylethanolamine; LacCer: lactosylceramide; DG: diacylglycero.

**Table 8 molecules-23-02929-t008:** Significant differentially-expressed proteins related to the neuroprotective effect of *N*-benzylhexadecanamide (XA).

NO.	Protein Name	Description	Ratio			Post-Model Trend
			M/K	XA50/M	XA100/M	
1	Aldoc	Fructose-bisphosphate aldolase C	0.68	2.30	2.33	↓
2	Dhfr	Dihydrofolate reductase	0.42	2.19	2.49	↓
3	Atp5d	ATP synthase subunit delta, mitochondrial	0.32	2.88	3.04	↓
4	Scarb2	Lysosome membrane protein 2	0.063	2.12	2.63	↓
5	Ak1	Adenylate kinase isoenzyme 1	0.13	1.62	2.23	↓
6	Vti1b	Vesicle transport through interaction with t-SNAREs homolog 1B	0	-	-	↓
7	Csnk2a2	Casein kinase II subunit alpha’	1.92	0.66	0.54	↑
8	Hmox1	Heme oxygenase 1	-	0.34	0	↑
9	Bnip2	BCL2/adenovirus E1B 19 kDa protein-interacting protein 2	1.89	0	0	↑
10	Mrpl50	39S ribosomal protein L50, mitochondrial	-	0	0	↑
11	Gla	Alpha-galactosidase A	-	0.69	0.61	↑

K: control group, M: model group, XA50:50 μM XA, XA100:100 μM XA.

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
