# Peer review of "Integrated Proteomics and Lipidomics Investigation of the Mechanism Underlying the Neuroprotective Effect of N-benzylhexadecanamide"

_molecules, 2018, doi:10.3390/molecules23112929_

Round 1

Reviewer 1 Report

The authors assessed N-benzylhexadecanamide (XA) a constituent of macamides, and showed the improved survival of neurons in vitro. The concentration of neurotransmitters in MN9D cells were used to investigate the neuroprotective effects of XA in 1-methyl-4-phenylpyridinium (MPP+)-induced neurodegeneration cell model. The authors indicate that differentially expressed biomarkers such as proteins and lipids may be responsible for the neuron-related activities of XA. Potential biomarkers related to the pathogenesis of neurodegenerative diseases after treatment with XA were assessed using an integrated proteomic and lipidomics approach. The protein–lipid interaction network indicate that the regulation of differentially expressed lipids/proteins determine sphingolipid metabolism and mitochondrial function  and underly the neuroprotectice effects of XA. The authors assessed various neurotransmitters such as taurine, norepinephrine, and the cholinergic constituents that correlated closely with the neuroprotective effects of XA. The authors indicate that their findings may improve the understanding of the multiple effects of macamides and their use in neurodegenerative diseases.

QUESTIONS:

Q1. Are the measurements of neurotransmitters to assess the neuroprotective effects of N-benzylhexadecanamide (XA) in 1-methyl-4-phenylpyridinium (MPP+)-induced neurodegeneration cell model the primary effects of XA and the integrated proteomic and lipidomic results secondary effects?

Q2. Are the lipidomic data derived from MN9D cells (dopaminergic neurons) associated with the various neurotransmitters with and without the effects of XA?

Q3. Can the multiple effects of macamides using XA as model compound for neurodegenerative diseases (mitophagy) be associated with metal binding effects (manganese) as the primary mechanism with effects on lipdomic as secondary effects?

Q4.  Are the effects of zinc and copper associated with neurodegenerative disease linked to the XA-manganese interaction in neurodegenerative diseases or can the XA effect zinc/copper versus copper/manganese versus zinc/manganese interactions or other metal interactions in the brain to prevent neurodegeneration?

RELEVANT REFERENCES:

Piomelli D1. The challenge of brain      lipidomics.Prostaglandins Other Lipid Mediat. 2005      Sep;77(1-4):23-34.

Gugnani KS1, Vu N2, Rondón-Ortiz AN1, Böhlke M1, Maher TJ1, Pino-Figueroa AJ3. Neuroprotective activity of      macamides on manganese-induced mitochondrial disruption in U-87 MG      glioblastoma cells. Toxicol      Appl Pharmacol. 2018 Feb 1;340:67-76.

Ross EE1, Hoag C2, Pfeifer Z2, Lundeen C2, Owens S2. Metal ion binding to phospholipid      bilayers evaluated by microaffinity chromatography. J Chromatogr A.      2016 Jun 17;1451:75-82.

Binder H1, Arnold K, Ulrich AS, Zschörnig O. Interaction of Zn2+ with phospholipid      membranes. Biophys Chem.      2001 Mar 15;90(1):57-74.

Corsetto PA1, Ferrara G2, Buratta S2, Urbanelli L2, Montorfano G1, Gambelunghe A3, Chiaradia E4, Magini A2, Roderi P1, Colombo I1, Rizzo AM5, Emiliani C6,7. Changes in Lipid Composition      During Manganese-Induced Apoptosis in PC12 Cells. Neurochem Res.      2016 Feb;41(1-2):258-69.

Courtney J. Mercadante, Carolina Herrera, Michael A. Pettiglio, Melanie L. Foster, Laura C. Johnson, David C. Dorman, and Thomas B. Bartnikas. The effect of high dose oral      manganese exposure on copper, iron and zinc levels in rats. Biometals. 2016 Jun; 29(3): 417–422.

Carl C.      Pfeiffer, and Scott LaMola, B.S. Zinc and Manganese in the Schizophrenias.      The Journal of Orthomolecular      Medicine Vol. 14, 1st Quarter 1999

Neggers YH1, Bindon JR, Dressler WW. The relationship between zinc and copper      status and lipid levels in African-Americans. Biol Trace      Elem Res. 2001 Jan;79(1):1-13.

Bowman AB1, Kwakye GF, Herrero Hernández E, Aschner M. Role      of manganese in neurodegenerative diseases. J Trace Elem Med Biol. 2011 Dec;25(4):191-203.

Henryk      Kozlowski , Marek Luczkowski , Maurizio Remelli , Daniela Valensin. Copper,      zinc and iron in neurodegenerative diseases (Alzheimer's, Parkinson's and      prion diseases). Coordination Chemistry Reviews. Volume 256, Issues 19–20,      October 2012, Pages 2129-2141

Author Response

Thank you very much for giving us an opportunity to revise our manuscript entitled “Integrated Proteomics and Lipidomics Investigation of the Mechanism Underlying the Neuroprotective Effect of N-benzylhexadecanamide” (381927) and deeply appreciate you and reviewer’s comments and suggestions. We have tried our best to improve our manuscript according to the comments. All the corrections and the responses to the reviewer’ comments were listed point by point below:

Reviewer #1 (Remarks to the Author):

Q1. Are the measurements of neurotransmitters to assess the neuroprotective effects of N-benzylhexadecanamide (XA) in 1-methyl-4-phenylpyridinium (MPP+)-induced neurodegeneration cell model the primary effects of XA and the integrated proteomic and lipidomic results secondary effects?

Response:Thanks for the reviewer’s comment. The measurements of neurotransmitters were the primary effects of XA. Thus, Measurement of Neurotransmitters were put in the pharmacodynamic evaluation part. The integrated proteomic and lipidomic study was mainly for the mechanism discussion.

Q2. Are the lipidomic data derived from MN9D cells (dopaminergic neurons) associated with the various neurotransmitters with and without the effects of XA?

Response:Thanks for the reviewer’s comment. According to network analysis based on HMDB and KEGG databases, our former experiment results showed that some differential neurotransmitters and lipids can be linked by some differential proteins. Among of which, the main differential proteins, such as Gla, Scarb2, Vti1b, Dhfr, Ak1, Csnk2a2, Mrpl50, can be used as key proteins for connection of differential neurotransmitters and lipids.

Q3. Can the multiple effects of macamides using XA as model compound for neurodegenerative diseases (mitophagy) be associated with metal binding effects (manganese) as the primary mechanism with effects on lipdomic as secondary effects?

Response:Thanks for the reviewer’s suggestive comment. As we all know, Manganese (Mn) and 1-methyl-4-phenylpyridinium (MPP+) both can cause a Parkinson-like syndrome. However, the mechanism may be different. 

The neurotoxin, 1-methyl-4-phenyl-1,2,3,6-tetrahydropyridine (MPTP) faithfully replicates most of the biological and pathological hallmarks in neurodegenerative diseases and was used to induce models of PD. Furthermore, MPTP has been crucial to the elucidation of mitochondrial dysfunction owing to their ability to inhibit mitochondrial electron transport complex activity, increase mitochondrial permeability transition, and reduce mitochonfrial movement. MPP+ was the active metabolite of the neurotoxin MPTP. MPP+ is then selectively absorbed into dopamine neurons where it exerts its neurotoxicity[1-5]. 

Manganese (Mn) is an essential trace element, which is necessary in physiological processes thatsupport development, growth, and neuron function. However, secondary to elevated exposure ordecreased excretion, Mn accumulates in basal ganglia in the brain and can cause a Parkinson-likesyndrome, referred to as manganism. Neuronal apoptosis plays an important role in the Mn neurotoxicity [6-8].

 What’s more, Manganese-Induced Apoptosis in PC12 Cells apoptosis is associated with changes in lipid composition detectable in whole cell extracts, namely cholesterol, phosphatidylserine and phosphatidylethanolamine decreases [9], which also indicating that the multiple effects of macamides using XA as model compound for neurodegenerative diseases (mitophagy) can be associated with metal binding effects (manganese) as the primary mechanism with effects on lipdomic as secondary effects. 

However, combined with our previous study [10], in this study we used MPP+ as the model compound. As for the effect of XA on manganese-induced neurodegerative diseases as the primary mechanism with effects on lipdomic as secondary effects , we will also discuss in our further study.

Q4.Are the effects of zinc and copper associated with neurodegenerative disease linked to the XA-manganese interaction in neurodegenerative diseases or can the XA effect zinc/copper versus copper/manganese versus zinc/manganese interactions or other metal interactions in the brain to prevent neurodegeneration?

Response:Many thanks for the review’s comment. The effects of zinc and copper can be associated with neurodegenerative disease linked to the XA-manganese interaction in neurodegenerative diseases. As the literatures reported[6], high dose oral manganese exposure impacted Cu, Fe and Zn levels in frontal cortex, a region known to exhibit neuronal degeneration and other pathologies following in vivo Mn exposure in non-human primates. Furthermore, macamides could alleviate the adverse effects of Mn on U-87 MG glioblastoma cells [11]. 

XA was the main ingredient of macamides. Thus, we conclude that the effects of zinc and copper can be associated with neurodegenerative disease linked to the XA-manganese interaction in neurodegenerative diseases, which will be our further research interest points.

We appreciate for reviewer’s warm work earnestly, and hope that our revision and response can meet the publication requirements of Molecules. Once again, thank you very much for your comments and suggestions.

Warmest regards.

Hai-Yu Zhao

Associate Professor, Ph.D.

Institute of Chinese Materia Medica, China Academy of Chinese Medical Sciences Beijing 100700, P.R. China

[1]N. Kanagaraj., H. Beiping., ST. Dheen.,SS. Tay. Downregulation of Mir-124 in MPTP-treated Mouse Model of Parkinson’s Diseases and MPP Iodide- Treated MN9D Cells Moduates the Expression of the Calpain/CDK5 pathway proteins. Neurosience 272 (2014): 167-169.

[2]Patil, SS., Jail, PD., Ghumatkar, PJ., Tambe, R., Sathaye, S. Neuroprotective effect of metformin in MPTP-induced Parkinson’s disease in mice. Neuroscience, 2014, 277:747-754.

[3] Kim, HG., Park, G., Piao, Y., Kang, MS., Park, YK., et al. Effects of the root bark of Paeonia suffruticosa on mitochondria-mediated neuroprotection in an MPTP-induced model of Parkinson’s disease. Food and Chemical Toxicology, 2014, 65: 293-300.

[4] Zhou, JJ., Zhai, SY., Zhang, HN., Wang, YH., Pu, XP. Neuroprotective effects of 3-O-demethylswertipunicoside against MPTP-induced Parkinson's disease in vivo and its antioxidant properties in vitro. Brain Research, 2015, 1624: 78-85.

[5] Lv, YQ., Zhang, Z., Hou, L., Zhang, L., Zhang, JY., et al. Phytic acid attenuates inflammatory responses and the levels of NFκB and p-ERK in MPTP-induced Parkinson’s disease model of mice. Neuroscience Letters, 2015, 597: 132-137.

[6] Mercadante, CJ., Herrera, C., Pettiglio, MA, et al.  The effect of high dose oral manganese exposure on copper, iron and zinc levels in rats. Biometals, 2016, 29(3): 1-6.

[7] Dobson, AW., Erikson, KM., & Aschner, M.Manganese neurotoxicity. Ann NY Acad Sci,2004, 1012:115–128.

[8] Guilarte, TR. Manganese and Parkinson’s disease: a critical review and new findings. Environ Health Perspect, 2010, 118:1071–1080

[9] Corsetto, PA., Ferrara, G., Buratta, S, et al. Changes in Lipid Composition During Manganese-Induced Apoptosis in PC12 Cells. Neurochem Res, 2016, 41(1-2): 258-269.

[10] Zhou, Y., Li, P., Brantner, A., Wang, H., Shu, X., & Yang, J., et al. Chemical profiling analysis of maca using UHPLC-ESI-Orbitrap MS coupled with UHPLC-ESI-QQQ MS and the neuroprotective study on its active ingredients. Sci Rep, 2017, 7: 44660.

[11] Gugnani, KS., Vu, N., Rondón-Ortiz, AN., Böhlke, M., Maher, TJ., & Pino-Figueroa, A. Neuroprotective activity of macamides on manganese-induced mitochondrial disruption in U-87 MG gliblastoma cells. Toxicol Appl Pharm, 2018, 340: 67-76.

Reviewer 2 Report

The authors present an interesting manuscript regarding the role of N-benzylhexadecanamide in the neuroprotective protection. I have some questions and doubts that I include as comments directly in the manuscript.

Author Response

Dear Editors and Reviewer:

Thank you very much for giving us an opportunity to revise our manuscript entitled “Integrated Proteomics and Lipidomics Investigation of the Mechanism Underlying the Neuroprotective Effect of N-benzylhexadecanamide” (381927) and deeply appreciate you and reviewer’s comments and suggestions. We have tried our best to improve our manuscript according to the comments. All the corrections and the responses to the reviewer’ comments were listed point by point below:

Q1. Line 42: Why do you write this reference in this way?

Response:Many thanks for the reviewer’s comment. It is my mistake. I have modified the reference in the revised manuscript.

Q2. Lline 54: reference?

Response:Thanks for the reviewer’s suggestion. I have added the reference [10], which is reference [7] in our revised manuscript.

Q3. line 183-187: Did you performed FASP digestion to remove the urea before trypsin disgetion? If not, how did you change the lysis buffer?

Response:Thanks for the reviewer’s comment. We did not do FASP digestion. Before trypsin disgestion, the urea was reduced to 1M by 50 mM ammonium bicarbonate (pH 8.0). we also presented it in the revised manuscript.

Q4. Line 216: Please, check if this value is correct. Are you using a Top50 method?

Response:Thanks for the reviewer’s comment. The value is correct, we used a Top50 method.

Q5. Line 219: Discoverer

Response:Thanks for the reviewer’s supportive suggestion. I have changed “Discovery” to “Discoverer”.

Q6. Did the authors include any modification in the peptide search, such as oxidation of methionine or carbamidomethylation of cysteine?

Response:Thanks for the reviewer’s comment. We include some modifications in the peptide search. The sentence “Oxidation (M) were chosen as variable modifications; carbamidomethylation (C) as fixed modification, two missed cleavage sites for trypsin was allowed.” was added in our revised manuscript.

Q7. Line 225, Change "expressed" by "regulated".

Response:We agreed with the reviewer’s comment. I have changed “expressed” by “regulated’.

Q8.Line 225: Why this p value?

Response:Thanks for the reviewer’s comment. It my mistake, the sentenced should be: “(with P values less than 0.01)”

Q9. Line 246: The.

Response:We agreed with the reviewer’s comment. I have modified this word.

Q10.Line 254: Could it be bigger?

Response:Thanks for the reviewer’s comment. I have made Figure 2 bigger.

Q11.Line 275: What is the meaning of S and C in the Figures S1 and S2?

Response:Thanks for the reviewer’s comment. “S” means cell supernatant. “C” means cells.

Q12. What about the rest of neurothansmitters measured? XA treatments increased aspartic acid, asparagine, serine and others in a different way that model treatment. Also, effects in supernatant and cells are not always the same but opposite for some neurotransmitters. Have tha authors any explanation for that? Could be relevant for the results? I think authors should discuss also these results.

Response:Thanks for the reviewer’s suggestion. According to our results, the concentration of taurine, norepinephrine, choline, acetylcholine, butyrylcholine, homovanillic acid, proline, histamine, serotonin, adrenaline, and levodopa displayed the same trend of variation in a dose-dependent manner in both the cell supernatant and MN9D cells. Among of them, the focal neurotransmitters including taurine, norepinephrine and the cholinergic neurotransmitters were addressed exhaustively. However, XA showed little adjustment for the rest of the neurotransmitters. Among of them, the contents of some neurotransmitters were very low and had no significant difference. Also, some neurotransmitters were not the predominant neurotransmitters that XA affected. 

    The sentence “However, XA showed little adjustment for the rest of the neurotransmitters.” was put in our revised manuscript.

Q13. Table 6: The column with the analyte names is missing.

Response:I have provided the analyte names in Table 6. 

Q14. Authors have done a lipidomic analyis of the samples and have the data about the modulation of specific PC, PE, LysoPC, and Cer. Since it have been reported different roles of these lipids in neurodegenation, at least for Cer (very long chain vs. long chain, etc, for instance: Neuromolecular Med. 2010 Dec;12(4):341-50. doi: 10.1007/s12017-010-8114-x. Epub 2010 May 26.), I miss some discussion about the potencial meaning of the modulation of the specific lipids, besides they as a group.

Response:Thanks for the reviewer’s constructive suggestion. We have discussed one specific lipid of different groups as an example in our revised manuscript. The discussion was shown as follows “Take PC(0:0/18:0) as an example, breviscapine ameliorated the learning and memory deficits of AD mice predominantly by regulating phospholipids metabolism. Among of them, PC(0:0/18:0) was one of potential biomarkers[12]; As for LysoPC(O-18:0), exposure to acephate disrupted metabolism of lipids, including lysoPC (15 : 0), lysoPC (16 : 0), lysoPC (O-18 : 0), lysoPC (18 : 1(9Z)), lysoPC (18 : 0), lysoPC(20 : 4(5Z, 8Z, 11Z, 14Z)), which induced oxidative stress and caused neurotoxicity[13]; As literature reported[14], C16- and C18-ceramides were closely associated with neurodegeration. ”

Q15. Figure 3, 4, 5, 6: Could it be bigger? 

Response:Thanks for the reviewer’s comment. I have made Figure 3, 4, 5, 6 bigger.

Q16.Please, include PC, PE, LacCer and DG abbreviation meaning.

Response:Thanks for the reviewer’s comment. PC means phosphatidylcholine, PE means phosphatidylethanolamine, LacCer means lactosylceramide, DG means diacylglycerol, which were also shown in revised manuscript.

Q17.Line 334: regulated

The term "expressed" is more used for gene regulation

Response:Thanks for the reviewer’s constructive comment. I have changed “expressed” by “regulated”.

Q18.Maybe authors can include the complete list of proteins altered by treatments as suplemmentary table (for example, in excel format).

Response:Thanks for the reviewer’s suggestive comment. I have put the complete list of proteins altered by XA treatment in Table S2 in our revised supplementary material.

Q19.Line 343-345: Full name of proteins because it is the first time that you cited.

Response:Thanks for the reviewer’s comment. The sentenced has been revised as “The expression of Fructose-bisphosphate aldolase C Aldoc, Dihydrofolate reductase (Dhfr),ATP synthase subunit delta, mitochondrial (Atp5d),Lysosome membrane protein 2 (Scarb2),Adenylate kinase isoenzyme 1(Ak1), and Vesicle transport through interaction with t-SNAREs homolog 1B (Vti1b) was decreased in the MPP+-induced cell model, whereas that of Casein kinase II subunit alpha’(Csnk2a2), Heme oxygenase 1(Hmox1), BCL2/adenovirus E1B 19 kDa protein-interacting protein 2(Bnip2), 39S ribosomal protein L50, mitochondrial (Mrpl50), and Alpha-galactosidase A (Gla) was increased.”

We appreciate for reviewer’s warm work earnestly, and hope that our revision and response can meet the publication requirements of Molecules. Once again, thank you very much for your comments and suggestions.

Warmest regards.

Hai-Yu Zhao

Associate Professor, Ph.D.

Institute of Chinese Materia Medica, China Academy of Chinese Medical Sciences Beijing 100700, P.R. China

[12] Xia, H., Wu, L., Chu, M. et al. Effects of breviscapine on amyloid beta 1-42 induced Alzheimer's disease mice: A HPLC-QTOF-MS based plasma metabonomics study. J Chromatogr B Analyt Technol Biomed Life Sci, 2017, 1057: 92-100.

[13] Hou, Y., Cao, C., Bao, W. et al. The plasma metabolic profiling of chronic acephate exposure in rats via an ultra-performance liquid chromatography-mass spectrometry based metabonomic method. Mol Biosyst, 2015, 11(2): 506-515

[14] Ben-David, O., &Futerman, AH. The role of the ceramide acyl chain length in neurodegeneration: involvement of ceramide synthases. Neuromolecular Med, 2010, 12(4): 341-350.

Reviewer 3 Report

The manuscript “Integrated Proteomics and lipidomic investigation of the mechanism underlying the

 Neuroprotective Effect of N-benzylhexadecanamide” was aimed to emphasize the integrated proteomics and lipidomics approach to investigate the mechanism underlying the neuroprotective effects of XA in an MPP+-induced neurodegeneration cell model.

On the basis of these results the Authors suggested that N-benzylhexadecanamide (XA), may be able to improve the survival rate of neurons in vitro. Finally the Authors concluded that the regulation of sphingolipid metabolism and mitochondrial function may be relevant for neuroprotection.

The study should be interesting in the field of new biomarker in neuroprotective effects.

However considering that toxic effects of MPTP on neuronal cells have been associated to mitochondrial dysfunction some biomarkers of oxidative stress should be considered. A sentence should be added to consideration of protective effects.

The sentence “Based on the Q-TOF MS and Orbitrap MS technology, lipidomic and proteomic analyses of the effects of XA on MPP+-induced MN9D cells were carried out” should be revised.

Author Response

Dear Editors and Reviewer:

Thank you very much for giving us an opportunity to revise our manuscript entitled “Integrated Proteomics and Lipidomics Investigation of the Mechanism Underlying the Neuroprotective Effect of N-benzylhexadecanamide” (381927) and deeply appreciate you and reviewer’s comments and suggestions. We have tried our best to improve our manuscript according to the comments. All the corrections and the responses to the reviewer’ comments were listed point by point below:

Q1. However considering that toxic effects of MPTP on neuronal cells have been associated to mitochondrial dysfunction some biomarkers of oxidative stress should be considered. A sentence should be added to consideration of protective effects.

Response:Thanks for the reviewer’s comment. We strongly agree with the reviewers' opinions. We have discussed in our revised manuscript as “MPP+ is reportedly mediated through oxidative mechanisms by inhibiting NADH-CoQ10 reductase (complex I) of the respiratory chain in mitochondria and then generating reactive oxygen species (ROS) [15-16]. It is well established fact that oxidative stress mechanism gets more prominent in the process of aging and the development of neurodegenerative disorders[17-18]. Our results demonstrated for the frst time that the neuroprotective effects of XA were accompanied by an improvement of mitochondrial respiratory function. Nevertheless, further studies are required to clarify the cause and effect relationship between reduction of oxidative stress and improvement of mitochondrial function by XA.”

Q2. The sentence “Based on the Q-TOF MS and Orbitrap MS technology, lipidomic and proteomic analyses of the effects of XA on MPP+-induced MN9D cells were carried out” should be revised.

Response:Thanks for the reviewer’s comment. Based on the Q-TOF MS and Orbitrap MS technology, the lipidomic and proteomic study of XA on MPP+-induced MN9D cells were carried out.

We appreciate for reviewer’s warm work earnestly, and hope that our revision and response can meet the publication requirements of Molecules. Once again, thank you very much for your comments and suggestions.

Warmest regards.

Hai-Yu Zhao

Associate Professor, Ph.D.

Institute of Chinese Materia Medica, China Academy of Chinese Medical Sciences Beijing 100700, P.R. China

[15] Javitch, JA., D’mato, RJ., Strittmatter, SM., Snyder, SH. Parkinsonism-inducing neurotoxin, MPTP: uptake of the metabolite MPP by dopamine neurons explains selective toxicity. Proc. Natl. Acad. Sci, 1985, 82: 2173-2177.

[16] Bates, TE., Heales, SJR., Davies, SEC., Boakye, P., Clark, JB. Effects of 1-methyl-4-phenylpyridium on isolated rat brain mitochondria: evidence for a primary involvement of energy depletion. J. Neurochem. 1994, 63: 640-648.

[17] Cui, H., Kong, Y., & Zhang, H. Oxidative stress, mitochondrial dysfunction, and aging. J Signal Transduct, 2012, doi: 10.1155/2012/646354. 

[18] Perier, C., & Vila, M. Mitochondrial Biology and Parkinson’s  disease. Cold Spring Harb Perspect Med, 2012, 4:1–19. 

Round 2

Reviewer 1 Report

RE: Manuscript “Integrated Proteomics and Lipidomics Investigation of the Mechanism Underlying the Neuroprotective Effect of N-benzylhexadecanamide

Thank you for the invitation to contribute to reviewing and to maintain high standards for peer-reveiwed journals. 

The authors have answered the questions raised by the reveiwer and indicated that these queries may be relevant to future prospects or studies.

The revised manuscript has appropriate revisions and responses and is now acceptable for scholarly publication.